Review

 

Subject Area:
cellular biology/biochemistry

Keywords:
microRNAs, colorectal cancer, biomarkers, diagnosis and prognosis

Author for correspondence:
Shufang Liang
e-mail: zizi2006@scu.edu.cn

†These authors contributed equally to this review.

# Emerging microRNA biomarkers for colorectal cancer diagnosis and prognosis

Bing Chen[1,2,†], Zijing Xia[1,3,†], Ya-Nan Deng[1], Yanfang Yang[1], Peng Zhang[4], Hongxia Zhu[5], Ningzhi Xu[1,5] and Shufang Liang[1]

[1]State Key Laboratory of Biotherapy and Cancer Center, West China Hospital, Sichuan University and Collaborative Innovation Center for Biotherapy, No. 17, 3rd Section of People's South Road, Chengdu 610041, People's Republic of China
[2]Department of Gastroenterology, The First Affiliated Hospital of Zhengzhou University, 1 Jianshe Eastern Road, Zhengzhou 450052, People's Republic of China
[3]Department of Nephrology, West China Hospital, Sichuan University, Chengdu 610041, Sichuan, People's Republic of China
[4]Department of Urinary Surgery, West China Hospital, West China Medical School, Sichuan University, Chengdu 610041, People's Republic of China
[5]Laboratory of Cell and Molecular Biology and State Key Laboratory of Molecular Oncology, Cancer Institute and Cancer Hospital, Chinese Academy of Medical Sciences, Beijing 100034, People's Republic of China

BC, 0000-0001-8188-3594; Y-ND, 0000-0002-4964-3483; YY, 0000-0001-7317-5651; SL, 0000-0003-1000-7508

MicroRNAs (miRNAs) are one abundant class of small, endogenous non-coding RNAs, which regulate various biological processes by inhibiting expression of target genes. miRNAs have important functional roles in carcinogenesis and development of colorectal cancer (CRC), and emerging evidence has indicated the feasibility of miRNAs as robust cancer biomarkers. This review summarizes the progress in miRNA-related research, including study of its oncogene or tumour-suppressor roles and the advantages of miRNA biomarkers for CRC diagnosis, treatment and recurrence prediction. Along with analytical technique improvements in miRNA research, use of the emerging extracellular miRNAs is feasible for CRC diagnosis and prognosis.

## 1. Introduction

Colorectal cancer (CRC) is one of the most commonly diagnosed cancers and a leading cause of death worldwide [1–3]. One of the important factors predicting survival in CRC patients is metastasis. In contrast to patients with stage II (no lymph node metastases) disease, the 5-year survival for patients with stage III CRC (with the presence of cancer cells in tumour-draining lymph nodes) declines more than 20% [4]. Nowadays, patients are diagnosed with CRC with a trend toward younger age [5]. In the last decade, CRC incidence rates increased by 22% and CRC death rates increased by 13% among adults aged less than 50 years in the USA [6]. However, the precise aetiologic factors of these onset cases have yet to be elucidated. As CRC develops slowly from removable precancerous lesions, early screening can reduce the incidence and mortality of this malignancy, and relevant diagnosis or prognosis biomarkers will be helpful for assessing tumour initiation, progression and response to treatment in CRC [7,8].

MicroRNAs (miRNAs) are small non-coding RNAs with 20–22 nucleotides, which participate in multiple biological processes [9]. miRNAs have been reported to be involved in occurrence and progression of various types of cancers, including brain, lung, breast, liver, prostate and colorectal cancer [10–16]. By targeting the 3′UTR of target genes, miRNAs influence a number of aspects of cancer cells by functioning as tumour suppressors or oncogenes, including mediating cell proliferation and migration [12,15], autophagy [10,14], apoptosis [11], metabolic shift [17], epithelial–mesenchymal transition [13,18] and radiosensitivity [19].

**Table 1.** Summary of dysregulated miRNAs in CRC.

| miRNA | dysregulation | target gene | effects | ref. |
|---|---|---|---|---|
| miR-155 | downregulated | CTHRC1 | suppress cell proliferation, promote cell cycle arrest and apoptosis | [23] |
| miR-205-5p | downregulated | ZEB1 | inhibit epithelial to mesenchymal transition | [18,24] |
| miR-18a | downregulated | CDC42 | inhibit colorectal cancer cell growth and death | [25] |
| miR-7 | downregulated | YY1 | resensitization to fas/FasL-apoptosis | [26] |
| miR-192/215 | upregulated | SRPX2 | facilitates cell glycolysis | [27] |
| miR-19b-1 | downregulated | ACSL/SCD | inhibit invasion in colon cancer cells | [28] |
| miR-30a | downregulated | metadherin | inhibit cell migration and invasion | [29] |
| miR-744 | downregulated | Notch1 | inhibit cell proliferation and invasion | [30] |
| miR-383 | downregulated | PAX6 | inhibit cell proliferation and invasion | [31] |
| miR-1271 | downregulated | Capn4 | suppress cell proliferation and invasion | [32] |
| miR-186-5p | downregulated | ZEB1 | inhibit cell proliferation, metastasis and epithelial to mesenchymal transition | [33] |
| miR-511 | downregulated | HDGF | reduce cell proliferation and invasion | [34] |
| miR-374b | downregulated | LRH-1 | inhibit cell proliferation and invasion | [35] |
| miR-216a-3p | downregulated | COX-2 and ALOX5 | suppress cell proliferation | [36] |
| miR-1273g-3p | upregulated | CNR1 | promote cell proliferation, migration and invasion | [37] |
| miR-494 | upregulated | APC | promote cell growth | [38] |
| miR-598 | upregulated | INPP5E | promote cell proliferation and cell cycle progression | [39] |
| miR-17-3p | upregulated | Par4 | promote cell proliferation and reduce apoptosis | [40] |
| miR-106a | upregulated | PTEN | promote cell proliferation and reduce apoptosis | [41] |
| miR-221 | upregulated | TP53INP1 | promote cell proliferation and reduce apoptosis | [42] |
| miR-214 | downregulated | ATG12 | promote radiosensitivity by inhibiting IR-induced autophagy | [43] |
| miR-26a | upregulated | PDHX | inhibit glucose metabolism | [16] |

miRNAs have already entered into cancer clinics as promising biomarkers [8]. Dysregulation phenotypes of miRNAs have been associated with CRC [16,20–22]. The candidate miRNA biomarkers are usually screened through exploring differential expression profiling of miRNAs in CRC tissues compared with the paired neighbouring non-cancerous colorectal tissues.

In this review, we summarize the differential expressions of miRNAs and their functions in CRC, and miRNA roles in CRC diagnosis, treatment and recurrence prediction. Moreover, the emerging extracellular miRNAs are illustrated as CRC biomarkers, owing to improvements in the detection approaches for miRNAs.

## 2. Aberrant miRNA expressions and roles in CRC

Aberrant expressions of miRNAs and their roles in various biological processes have been revealed to be associated with CRC carcinogenesis. miRNAs function as either oncogenes or tumour suppressors by regulating different targets (table 1). For example, miR-18a [25], miR-155 [23,44] and miR-205-5p [18,24] repress proliferation, migration and invasion of CRC cells, whereas miR-494 [38], miR-598 [39] and miR-17-3p [40] promote the abilities of cell proliferation, migration and invasion. miR-106a [41] and miR-7 [26] relate to apoptosis of CRC cells or the resistance to apoptosis. miR-221 and miR-214 reduce autophagy in CRC cells [42,43]. miR-192/215 [27] and miR-19b-1 [28] have regulatory

roles in metabolic pathways. Other miRNAs regulating different targets are also included in table 1 [29–37].

miR-508 induces the stem-like/mesenchymal subtype in CRC by affecting the expression of cadherin CDH1 and the transcription factors ZEB1, SALL4 and BMI1 [45]. miR-15A and 16-1 reduce the chemotaxis of IgA(+) B cells and activate signalling pathways required for B-cell-mediated immune suppression in CRC [46]. miR-21-5p exhibits epigenetic effects in CRC by blocking the activation of DNA demethylation [47].

Owing to the high-throughput genome-wide profiling and comprehensive screening technologies, novel miRNA signatures [48,49] and multiple miRNA–mRNA regulatory networks [50,51] have been discovered in CRC. miRNA-mediated genes and signalling pathways are also explored for their associations with CRC. NF-κB regulates immune response and inflammation processes, and is associated with multiple miRNAs such as miR-150-5p, miR-195-5p and miR-203a in carcinogenesis [52]. These studies indicate miRNAs and their targets of action are tightly involved in CRC progression, which gives a promising outlook for the identification of miRNA biomarkers for CRC.

## 3. miRNAs for CRC diagnosis, treatment and recurrence prediction

miRNAs have shown great clinical value in the diagnosis, treatment and prognosis of CRC (figure 1). Developing appropriate miRNA biomarkers is essential for early stage CRC diagnosis. The differential miRNAs and miRNA-regulated genes were screened in early stage CRC tissues,

royalsocietypublishing.org/journal/rsob Open Biol. **9**: 180212

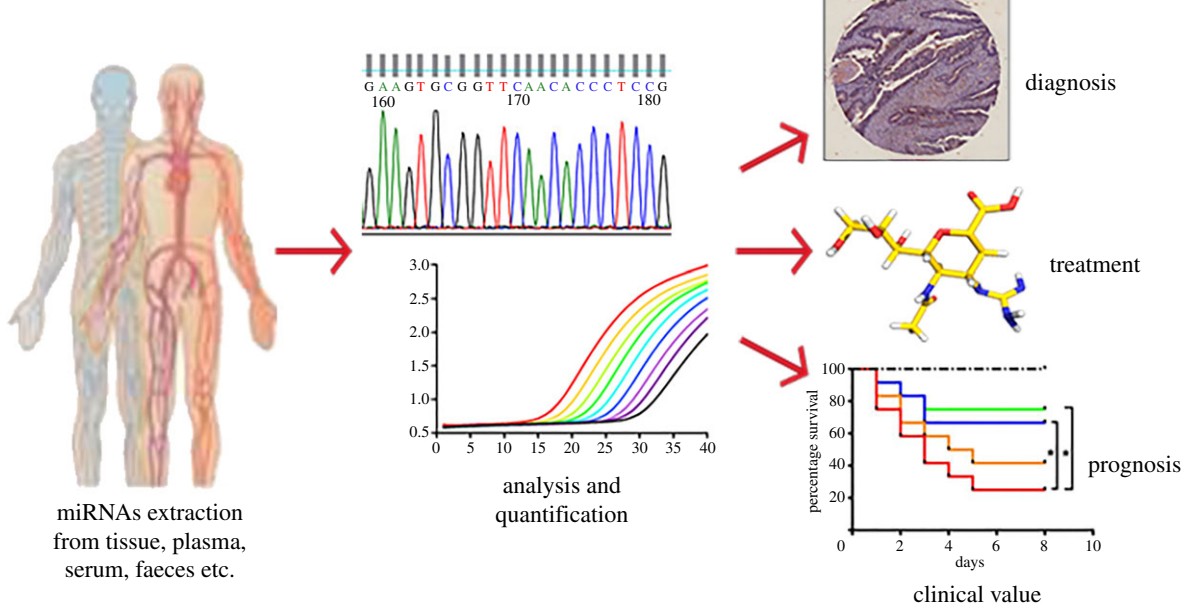

**Figure 1.** Promising clinical value of miRNAs in diagnosis, treatment and prognosis of CRC. Analysing and quantifying the levels of miRNAs extracted from clinical samples such as tissue, plasma, serum and faeces can help to diagnose and make decisions for CRC treatment and prognosis.

precancerous lesions and colonic intraepithelial neoplasia by RNA sequencing [53]. miR-548c-5p, miR-548i and miR-548am-5p were identified as the top three of 26 differentially expressed miRNAs with regard to lymph node metastasis [53]. But some other studies report no significant correlations of miRNA-expression patterns with CRC tumour stage by a miRNA microarray analysis [54]. The difference in conclusions is probably due to different detection approaches and tumour sample heterogeneity.

Through bioinformatics analysis from the Cancer Genome Atlas, five significantly changed miRNAs (miR-32, miR-181b, miR-193b, miR-195 and miR-411) were uncovered in T1 and T2 CRCs with versus without lymph node metastases [55]. This five-miRNA signature was validated with a greater degree of accuracy in two cohorts of patients with T1 cancers and untreated patients [55]. In addition, miRNAs are correlated with molecular histological markers (such as Ki-67 and CD34), which is helpful to determine cell proliferation and angiogenesis in CRC development [56]. New efforts have been made to identify differential expression patterns of miRNAs in genotype-specific CRC. Significantly higher levels of miR-31 and obvious lower miR-373 is present in BRAF-mutated tumours compared with wild-type tumours [57]. Regrettably, there was no difference in expression levels between KRAS- and BRAF-mutated tumours in this study. However, it was a good attempt to investigate miRNA signatures for clinico-pathological differences between KRAS- and BRAF-mutated CRCs.

Furthermore, miRNAs have great potential in CRC treatment and could help to overcome the resistance to cancer therapy. For instance, miR-214 enhances CRC radiosensitivity by inhibiting autophagy in CRC cells [43]. The overexpression of miR-143 is related to the oxidative stress and cell death in CRC cells, which might circumvent resistance of CRC cells to oxaliplatin [58]. miR-195 is able to desensitize CRC cells to 5-fluorouracil (5-FU) [59,60]. Meanwhile, a miR-129 mimic is reported to enhance efficacy to eliminate resistance to 5-FU in CRC stem cells [61]. It is notable that the miR-129 mimic can be delivered to cancer cells without any transfection reagents, including lipids, viral vectors and nanoparticles. Another trial used miR-20a-loading

nanoparticles to target liver sinusoidal endothelial cells, and showed a great reduction of liver metastasis of CRC both *in vitro* and *in vivo* [62]. All the trials indicate promising miRNA-based therapy for CRC. Moreover, small molecules such as polyamine derivatives have also been developed as miRNA interfering agents to tackle the overexpression of oncogenic miRNAs [63].

So far, miRNAs are promising as biomarkers with prognostic and predictive values [64,65]. A recent study has confirmed an eight-miRNA signature is feasible for predicting tumour recurrence of CRC patients in stages II and III by using three independent genome-wide miRNA-expression profiling datasets [48]. In this study, miRNA biomarkers for CRC were screened from multiple clinical cohorts totalling 736 patients, including patients from the publicly available dataset and two clinical validation cohorts of CRC patients who underwent surgery without neoadjuvant chemotherapy. From 25 miRNAs identified, eight candidates were selected with top statistical significance ($p$-value < 0.2), including hsa-mir-191, hsa-mir-200b, hsa-mir-30b, hsa-mir-30c2, hsa-mir-33a, hsa-mir-362, hsa-mir-429 and hsa-mir-744, which were further validated in fresh frozen tissues [48]. Moreover, miRNAs from tumour-adjacent mucosa have been studied for the prediction of relapse risk after curative treatment. Four miRNAs (miR-18a, miR-21, miR-182 and miR-183) were found with coordinate deregulation in the normal mucosa adjacent to the tumour, which is predictive of relapse within 55 months from curative surgery in 48 localized CRC patients who underwent radical tumour resection [66].

## 4. Emerging extracellular miRNAs used for CRC biomarkers

miRNAs can be detected not only in tissue samples but also in extracellular samples including faeces and blood (table 2). miRNAs from different blood or faeces samples were useful for CRC diagnosis or prognosis [67–70,72–75,81]. miRNAs have been reported to be relatively stable in a variety of biological fluids and even formalin-fixed paraffin-embedded tissues

**Table 2.** Extracellular miRNA markers from blood or faecal specimen. AUC, area under the curve.

| source | miRNA | case (n) | control (n) | sensitivity (%) | specificity (%) | AUC (95% CI) | detection method | clinical use | ref. |
|---|---|---|---|---|---|---|---|---|---|
| plasma | miR-21 | 31 | 34 | 65 | 85 | — | qPCR | diagnosis | [67] |
| | miR-21 | 186 | 53 | 82.8 | 90.6 | 0.919 (0.87–0.96) | qPCR | diagnosis and prognosis | [68] |
| | miR-6826 | 93 | — | — | — | 3.670 | qPCR | prediction for a poor response to vaccine treatment | [69] |
| serum | miR-122 | 543 | — | — | — | — | array microRNA cards | prognosis | [70] |
| | miR-24-2 | 228 | 68 | — | — | — | qPCR | diagnosis | [71] |
| | miR-139-3p | 117 | 90 | 96.6 | 97.8 | 0.9935 | qPCR | diagnosis | [72] |
| | miR-139-5p | 53 | — | 64 | 80 | 0.59–0.87 | qPCR | prognosis | [73] |
| | miR-135a-5p | 60 | 40 | — | — | 0.832 (0.73–0.93) | qPCR | diagnosis | [74] |
| | | 60 | 50 | — | — | 0.875 (0.80–0.95) | | | |
| | miR-203 | 330 | — | — | — | — | qPCR | prognosis and metastasis prediction | [75] |
| | (miR-21 + miR-29 + miR-92 +miR-125 +miR-223) | 85 | 78 | 84.7 | 98.7 | 0.952 | qPCR | diagnosis | [76] |
| | miR-24 | 111 | 130 | 78.38 | 83.85 | 0.839 | qPCR | early detection | [77] |
| | miR-320a | | | 92.79 | 73.08 | 0.886 | | | |
| | miR-423-5p | | | 91.89 | 70.77 | 0.833 | | | |
| | (miR-24 + miR-320a +miR-423-5p) | | | 92.79 | 70.77 | 0.899 | | | |

(Continued.)

**Table 2.** (*Continued.*)

| source | miRNA | case (n) | control (n) | sensitivity (%) | specificity (%) | AUC (95% CI) | detection method | clinical use | ref. |
|---|---|---|---|---|---|---|---|---|---|
| plasma exosomes | miR-21 | 326 | — | — | — | — | TaqMan miRNA assays | prediction of recurrence and poor prognosis in CRC patients with TNM stage II, III or IV. | [78] |
| | miR-6803-5p | 168 | — | — | — | — | qPCR | diagnosis and prognosis | [79] |
| | miR-17-5p | 18 | 10 | — | — | 0.897 (0.80–0.99) | qPCR | prognosis | [80] |
| | | 11 | 10 | — | — | 0.841 (0.72–0.96) | | | |
| | miR-92a-3p | 18 | 10 | — | — | 0.845 (0.72–0.97) | | | |
| | | 11 | 10 | — | — | 0.854 (0.74–0.97) | | | |
| saliva | miR-21 | 34 | 34 | 97 | 91 | — | qPCR | diagnosis | [67] |
| faecal | miR-4478 | 40 | 16 | — | — | — | SYBR Green miScript PCR system | diagnosis | [81] |
| | miR-1295b-3p | 40 | 16 | — | — | — | | | |

royalsocietypublishing.org/journal/rsob Open Biol. 9: 180212

[82,83]. In view of this, miRNAs present as promising targets for non-invasive biomarker development in CRC and are attractive for eventual clinical translation. miRNAs such as miR-129 with a high expression level in CRC plasma [84] and miR-24-2 with a low level in CRC serum [71] can be potential positive or negative biomarkers in the diagnosis of CRC patients.

More studies were performed to profile novel circulating miRNA biomarkers for CRC. A pilot study identified five miRNAs, including miR-31, miR-141, miR-224-3p, miR-576-5p and miR-4669, to be significantly different in sera between patients with colon cancer and healthy controls, indicating a miRNA panel for diagnosis of CRC [85]. Faecal sampling is also an ideal non-invasive alternative for detection [86]. Several miRNAs, including miR-29a [87], miR-223 [88], miR-224, miR-106a [89] and miR-135b [90], are detectable in the faeces and could be informative biomarkers for screening and diagnosis of CRC. A faecal miRNA test combined with routine immunochemical faecal occult blood test (iFOBT) is helpful to identify CRC patients from those with negative iFOBT results and may improve the sensitivity to detect CRC [89].

More and more studies focus on circulating exosomal miRNAs [91–97]. Exosomes represent a type of extracellular vesicle formed from endosomal membrane with encapsulated cystolic contents, containing proteins, mRNAs and miRNAs [98,99]. miRNA-containing exosomes have been isolated from various body fluids, and through the exosomal pathway, miRNAs can be sampled from donor cells and transferred between cells [100]. Circulating exosomal miRNAs are considered as novel diagnostic and prognostic biomarkers for CRC [101,102]. For example, levels of miR-17-5p and miR-92a-3p isolated from serum exosomes were identified to associate with pathologic stages and grades of the CRC patients [80]. High levels of serum exosomal miR-6803-5p were observed in CRC patients, which are correlated with TNM stage (a staging system to describe the amount and spread of tumour in a patient's body, in which T describes the size of the original tumour and whether it has invaded nearby tissue, N describes nearby lymph nodes that are involved and M describes metastasis), lymph node metastasis, liver metastasis as well as poor overall survival and disease-free survival [79]. Serum exosomal miR-4772-3p is considered to be a prognostic biomarker for tumour recurrence in stages II and III CRC patients [103]. Also, miR-21 from serum exosomes is a useful biomarker for the prediction of CRC recurrence and poor prognosis with each tumour stage including TNM stages II, III or IV [78].

# 5. Advantages and disadvantages of miRNA biomarkers

Despite the great success in decreasing the incidence and mortality of CRC that colonoscopy has achieved [104,105], the process of examination is not friendly enough, being accompanied by discomfort, fear of pain and embarrassment [106,107]. At present, the commonly used plasma biomarkers in CRC detection are carcinoembryonic antigen (CEA) and carbohydrate antigen 19-9 (CA19-9), but the practical application of both is limited due to their low sensitivity and specificity [108,109]. Emerging evidence indicates that miRNA is a potential choice for CRC detection compared with traditional blood-based biomarkers such as CEA and CA19-9.

A publication evaluated the diagnostic efficiency of miRNAs and traditional biomarkers such as CEA and CA19-9 [76]. A total of 9936 CRC patients and 7935 healthy controls were included in this study, and the results showed that the concentrations of CEA and CA19-9 were both elevated in the CRC group and significantly different from the healthy control group ($p < 0.001$ and $p = 0.004$, respectively), indicating that CEA and CA19-9 were capable for CRC detection. However, the combination of five miRNAs (miR-21 + miR-29 + miR-92 + miR-125 + miR-223) showed a better sensitivity (84.7%) and specificity (98.7%) compared with CEA (69.4% sensitivity, 78.2% specificity) and CA19-9 (65.9% sensitivity, 67.1% specificity) by assessing the sensitivity, specificity, Youden index and the area under the curve (AUC) of the receiver operating curve through meta-analysis methods.

Several miRNAs are potential biomarkers for CRC detection. Through data analysis from 223 CRC patients and 130 healthy controls [77], the sensitivity of miR-24, miR-320a and miR-423-5p for early stage CRC were 77.78%, 90.74% and 88.89%, which were better than CEA and CA19-9 (40.54% and 36.04%, respectively). Moreover, the diagnostic efficiencies of miR-24, miR-320a and miR-423-5p were much higher than traditional CEA and CA19-9 (81.33%, 82.16%, 80.50% versus 70.12%, 66.80%). In addition, the combination of a few miRNAs has been investigated for potential detection of early stage CRC, including a five-serum miRNA detection panel (miR-1246, miR-202-3p, miR-21-3p, miR-1229-3p and miR-532-3p) [110] and a three-plasma miRNA detection panel [111]. In conclusion, miRNAs have been shown to be a powerful tool for CRC detection in a non-invasive manner and have better performance in CRC detection compared with conventional blood-based biomarkers such as CEA and CA19-9.

Even so, miRNA biomarkers still have a number of practical problems that need to be resolved. One of them is that establishment of a sensitive detection method suitable for blood samples is required. The cost is also a concern. On average US$23 is required for a four-miRNAs panel detection in China, while the detection of two known cancer biomarkers, CEA and CA19-9, respectively, costs US$4.6 and US$8.0 for each sample [76].

Moreover, no single miRNA alone has been identified as an ideal CRC biomarker upto now. Similarly to other gene or protein cancer markers, some predictive miRNAs are usually not specific for one kind of cancer. For example, miR-18a is reported to be a tumour suppressor by suppressing CDC42 in CRC [25]. However, miR-18a is also a candidate biomarker for breast cancer and lung cancer, having a significantly higher expression in benign breast biopsy than normal controls [112] and correlating with poor prognosis in patients with non-small cell lung cancer [113]. Similarly, miR-155 inhibits colorectal cancer progression and metastasis [23], while it is significantly overexpressed in breast cancer and cervical cancer with potential as a biomarker [114,115]. Fortunately, a panel of miRNAs can be used to distinguish CRC patients from healthy controls with a relatively high sensitivity and specificity, from testing in a large population of subjects [116]. Therefore, several miRNA combinations are feasible to monitor cancer profiling.

# 6. Regulation and detection of miRNAs

miRNAs represent a type of approximately 22-nucleotide non-coding RNA molecule. The biogenesis and maturation of miRNAs comprise several regulated steps [117,118]. For the

mechanism of aberrant miRNA-expression levels, miRNA processing was taken into consideration, involving various regulatory proteins (such as Drosha [119] and Dicer [120]) and cellular location [121]. In addition, genetic alterations, single nucleotide polymorphisms [122] and epigenetic modification (such as DNA methylation [123,124] and N6-methyladenosine [125]) can also affect the processing efficiency of miRNAs. In terms of the extracellular miRNAs, they are subsequently packed in apoptotic bodies, microvesicles and exosomes or bound to RNA-binding proteins, and then released from donor cells [126–128]. From this aspect, factors involved in the secretion processes can affect the abundance of extracellular miRNAs, including triggering secretion of exosomes [129], exosome biogenesis [130] and membrane trafficking [131].

The current gold standard for detecting precancerous adenomas and colorectal cancers is still colonoscopy, but its invasiveness puts patients in great pain. Therefore, miRNA as a non-invasive option has attracted a lot of attention. If miRNAs are to be a reliable and convenient biomarker for the detection and prognosis of CRC, detection methods with high sensitivity and specificity, but also practicality, operability and low price, are indispensable. Conventional detection methods for miRNAs are qPCR, microarray and next-generation sequencing, but no one method is completely ideal for clinical application. In order to meet clinical requirements, various methods such as isothermal amplification techniques and near-infrared technology have been established and have been summarized elsewhere [132].

Here, we focus on several newly reported methods. A method based on ionic liquid (IL; 1-butyl-3-methylimidazolium hexafluorophosphate)-modified chemically activated pencil graphite electrodes (IL/CA/PGEs) [133] has been developed for miR-34a detection, with the advantages of being easy-to-handle, cost-effective, fast and portable. In this platform, the detection limits of miRNA achieved were 109 nM in PBS buffer and 117 nM in diluted FBS medium. The method also shows a good selectivity against other miRNAs. However, its performance with real whole blood has not been characterized, because whole blood detection is an arduous task owing to the existence of interfering factors.

Another energy-transfer-based photoelectrochemical method was developed to detect miR-141 with an integration of entropy-driven toehold-mediated DNA strand displacement (ETSD) reaction with magnetic beads (MBs) [134]. This system improved the sensitivity significantly compared to the absorption and photoluminescence method by elevating the detection limit to 0.5 fM. This protocol was also performed using real blood samples and the average recovery of miR-141 increased from 96 to 108%, which is an acceptable accuracy and reproducibility. Two other published methods, an enzyme-free miRNA target-triggered strand displacement reaction (SDR) amplification strategy-based biosensor [135] and an inductively coupled plasma mass spectrometry (ICP-MS)-based strategy [136], were used for detecting miR-21. In the SDR amplification strategy-based biosensor method, miR-21 was coupled with the redox signal of ferrocene and the detection limit was decreased to 0.34 fM with a good selectivity and reproductivity. Even in real serum samples, it worked well with a recovery value ranging from 92 to 113% [135]. The ICP-MS-based strategy was also a miRNA-triggered system through detection of the isotope $^{89}$Y. Target miRNA triggers a chain reaction for alternating hybridization between DNA H1 (bond on UCNPs@DNA probe) and DNA H2, leading to accumulation

of ultra-small lanthanide upconversion nanoparticles (UCNPs), which is correlated with the concentration of miR-21. This method provided an alternative option for detecting miR-21 in serum with a detection limit of 41aM [136].

Unlike the two methods above, the third one is a way of direct detection of the target using a tool named molecular beacons [137]. It therefore possesses the advantages of being direct, simple and rapid. However, the sensitivity of the molecular beacons based method is worse owing to the lack of signal amplification. An amplification-free electrochemical method shows good detection sensitivity for exosomal miR-21 from serum samples [138]. The target miRNAs are selectively enriched and isolated through MBs, which are pre-functionalized with capture probes to directly adsorb the targets onto a gold electrode surface. The adsorbed miRNAs are measured electrochemically in the presence of an $(Fe(CN)_6)4-/3-$ redox system [138].

In order to improve the quantification of extracellular miRNAs, an optimized protocol has been reported to assess a panel of up to 20 miRNAs in serum samples by using multiplexed Taqman miRNA stem-loop primers in the reverse transcription step [139]. Extra isolation steps for pure exosomes are requisite for detection of exosomal miRNAs, including solely or jointly using ultracentrifugation, size-based isolation, immune-affinity capture, microfluidics-based platforms and water excluding polymer-based methods, depending on structural features of the exosomes [140]. To simplify the procedure of miRNA detection, a detection method without miRNA purification or labelling was provided by Fujii *et al.* [141]. Collecting extracellular vesicles is not required in this system. Prior to loading a sample, heating is performed at 98°C for 2 min to collapse the extracellular vesicle for the release of miRNA, no matter what kind of sample, blood or urine. Despite its convenience, the downside of this system is insufficient sensitivity, which requires improvement for practical diagnosis. The sensitivity of another purification-free method is, however, better; this is a fishhook probe-based rolling circle amplification (FP-RCA) assay [142]. The linear relationship of FP-RCA showed a good correlation at miRNA concentrations from 100 fM to 10 pM. In addition, the FP-RCA assay is of simplicity, low cost and portability.

# 7. Perspective

So far, miRNAs have shown a strong potential for their utilization as oncological biomarkers in CRC. Direct non-invasive detection of circulating miRNAs would provide information for diagnosis, prognosis and predictive treatment responses for CRC patients. However, the feasibility of exosomal miRNAs as non-invasive biomarkers in CRC still remains a concern, because of multiple tissue sources of a circulating miRNA. Moreover various confounding socioeconomic, environmental and lifestyle factors have profound effects on miRNA levels [143,144]. To realize the potential of miRNA biomarkers for use in personalized treatment, these factors deserve further exploration of their influences on the miRNA-expression patterns.

Data accessibility. This article has no additional data.

Authors' contributions. All authors participated in the preparation of the manuscript, and read and approved the final manuscript.

Competing interests. The authors declare that they have no competing interests.

**Funding.** This work was supported by the National 863 High Tech Foundation (2014AA020608), the National Key Basic Research Program of China (2013CB911303 and 2011CB910703), the National Natural Science Foundation of China (31470810), the Science & Technology Department of Sichuan Province (2017JY0232), the Health and Family Planning Commission of Sichuan Province (17ZD045) and Chengdu Science and Technology Program (2017-GH02-00062-HZ).

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
