## [Reviewer comments · Open Biology]

Review History

RSOB-18-0212.R0 (Original submission)

Review form: Reviewer 1

Recommendation

Accept with minor revision (please list in comments)

Are each of the following suitable for general readers?

- a) **Title**
Yes
- b) **Summary**
Yes
- c) **Introduction**
Yes

Is the length of the paper justified?

Yes

Should the paper be seen by a specialist statistical reviewer?

No

Is it clear how to make all supporting data available?

Yes

Is the supplementary material necessary; and if so is it adequate and clear?

Not Applicable

Do you have any ethical concerns with this paper?

No

Comments to the Author

The manuscript entitled "Emerging microRNA biomarkers for colorectal cancer diagnosis and prognosis" reviewed the latest progress of potential applications of microRNA as biomarkers for the diagnosis and prognosis of colorectal cancer (CRC). The authors also discussed the advantages of using microRNA as biomarkers and the technical progresses in microRNA detection. This manuscript is well organized. The topic and content are attractive, providing a very good literature resource for understanding the progress of microRNA research in cancer studies.

The manuscript may be accepted for publication in the journal when several issues are properly addressed or corrected.

1. It would be better if the disadvantages of using microRNA as biomarkers are also summarized and discussed in the section subtitled "Advantages of microRNA biomarkers" on page 8.
2. Is the list of microRNA biomarkers specific for CRC? Can any of them be applied to other types of cancers?
3. Some sentences are not clear. For example: On the line 7, page 5, the sentence "Regarding to lymph node metastasis, although a previous evaluation about the correlations of miRNA expression patterns with CRC tumor stage by using miRNA microarrays showed no significant differences". On the line 24, page 5, the sentence "Furthermore, miRNAs have great potential in CRC treatment and the overcome of resistance to cancer therapy" is difficult to understand.
4. Abbreviations should be consistency in the manuscript. For example, the miRNA and microRNA. On the line 14, page 8, the form of the letter "p" ($p < 0.001$ and $P = 0.004$).

Review form: Reviewer 2

Recommendation

Accept with minor revision (please list in comments)

Are each of the following suitable for general readers?

a) **Title**
Yes

b) **Summary**
Yes

c) **Introduction**

Yes

Is the length of the paper justified?

Yes

Should the paper be seen by a specialist statistical reviewer?

Yes

Is it clear how to make all supporting data available?

Not Applicable

Is the supplementary material necessary; and if so is it adequate and clear?

Not Applicable

Do you have any ethical concerns with this paper?

No

Comments to the Author

The reviewer appreciated the opportunity to be invited to read and revise the manuscript.

However, the reviewer just felt it would be much convenient to do the review if the editorial provides line numbers to the article, especially to such as a long review article.

Overall, the manuscript was well written, and the citations were comprehensive. Actually, all the references were checked manually to ensure correct citation. Minor revisions:

1. Page 3 (Introduction), 3rd paragraph: "as promising biomarkers 8." Need format revision.

2. Page 4 (Aberrant miRNA expressions and roles in CRC), 1st paragraph: please make sure [23] is the right reference; may include the MiR-7, 192/215 and 19b-1 into Table 1.

2nd paragraph: "... SALL4, BMI1 and BMI1 [36]". Please remove duplicates.

3. Page 5, 2nd paragraph: "Through bioinformatics analysis form the Cancer Genome Atlas", should be "from".

4. Page 7, 3rd paragraph: "Exosomes represent a kind of intracellular vesicles ...". Is it "extracellular"?

TNM: please define.

5. Page 8: ref [90]: please make sure it is the right ref.

The middle section (e.g., %, specificity, sensitivity) is confusing; should be clarified or moved to Table 2.

6. Page 9, 2nd paragraph: regarding the ref [90], please double-check the descriptions in this paragraph and ensure this is the right article to cite.

7. Page 11, 1st paragraph: "89Y?". Please clarify.

3rd paragraph: "immune-affinity" should be immune-affinity?

Table 1: "miR-205-5p" needs one more citation [26].

"miR-214": ref [32] is not appropriate.

Decision letter (RSOB-18-0212.R0)

17-Dec-2018

Dear Professor Liang,

We are pleased to inform you that your manuscript RSOB-18-0212 entitled "Emerging microRNA biomarkers for colorectal cancer diagnosis and prognosis" has been accepted by the Editor for

publication in Open Biology. The reviewer(s) have recommended publication, but also suggest some minor revisions to your manuscript. Therefore, we invite you to respond to the reviewer(s)' comments and revise your manuscript.

Please submit the revised version of your manuscript within 14 days. If you do not think you will be able to meet this date please let us know immediately and we can extend this deadline for you.

- 1) A text file of the manuscript (doc, txt, rtf or tex), including the references, tables (including captions) and figure captions. Please remove any tracked changes from the text before submission. PDF files are not an accepted format for the "Main Document".
- 2) A separate electronic file of each figure (tiff, EPS or print-quality PDF preferred). The format should be produced directly from original creation package, or original software format. Please note that PowerPoint files are not accepted.
- 3) Electronic supplementary material: this should be contained in a separate file from the main text and meet our ESM criteria (see <http://royalsocietypublishing.org/instructions-authors#question5>). All supplementary materials accompanying an accepted article will be treated as in their final form. They will be published alongside the paper on the journal website and posted on the online figshare repository. Files on figshare will be made available approximately one week before the accompanying article so that the supplementary material can be attributed a unique DOI.

Online supplementary material will also carry the title and description provided during submission, so please ensure these are accurate and informative. Note that the Royal Society will not edit or typeset supplementary material and it will be hosted as provided. Please ensure that the supplementary material includes the paper details (authors, title, journal name, article DOI). Your article DOI will be 10.1098/rsob.2016[last 4 digits of e.g. 10.1098/rsob.20160049].

- 4) A media summary: a short non-technical summary (up to 100 words) of the key findings/importance of your manuscript. Please try to write in simple English, avoid jargon, explain the importance of the topic, outline the main implications and describe why this topic is newsworthy.

Images

Data-Sharing

It is a condition of publication that data supporting your paper are made available. Data should be made available either in the electronic supplementary material or through an appropriate repository. Details of how to access data should be included in your paper. Please see <http://royalsocietypublishing.org/site/authors/policy.xhtml#question6> for more details.

Data accessibility section

Sincerely,

The Open Biology Team
<mailto:openbiology@royalsociety.org>

Reviewer(s)' Comments to Author:

Referee: 1

Comments to the Author(s)

The manuscript entitled "Emerging microRNA biomarkers for colorectal cancer diagnosis and prognosis" reviewed the latest progress of potential applications of microRNA as biomarkers for the diagnosis and prognosis of colorectal cancer (CRC). The authors also discussed the advantages of using microRNA as biomarkers and the technical progresses in microRNA detection. This manuscript is well organized. The topic and content are attractive, providing a very good literature resource for understanding the progress of microRNA research in cancer studies.

The manuscript may be accepted for publication in the journal when several issues are properly addressed or corrected.

1. It would be better if the disadvantages of using microRNA as biomarkers are also summarized and discussed in the section subtitled "Advantages of microRNA biomarkers" on page 8.
2. Is the list of microRNA biomarkers specific for CRC? Can any of them be applied to other types of cancers?
3. Some sentences are not clear. For example: On the line 7, page 5, the sentence "Regarding to lymph node metastasis, although a previous evaluation about the correlations of miRNA expression patterns with CRC tumor stage by using miRNA microarrays showed no significant differences". On the line 24, page 5, the sentence "Furthermore, miRNAs have great potential in CRC treatment and the overcome of resistance to cancer therapy" is difficult to understand.

4. Abbreviations should be consistency in the manuscript. For example, the miRNA and microRNA. On the line 14, page 8, the form of the letter “p” (p < 0.001 and P = 0.004).

Referee: 2

Comments to the Author(s)

The reviewer appreciated the opportunity to be invited to read and revise the manuscript.

However, the reviewer just felt it would be much convenient to do the review if the editorial provides line numbers to the article, especially to such as a long review article.

Overall, the manuscript was well written, and the citations were comprehensive. Actually, all the references were checked manually to ensure correct citation. Minor revisions:

1. Page 3 (Introduction), 3rd paragraph: “as promising biomarkers 8.” Need format revision.

2. Page 4 (Aberrant miRNA expressions and roles in CRC), 1st paragraph: please make sure [23] is the right reference; may include the MiR-7, 192/215 and 19b-1 into Table 1.

2nd paragraph: “... SALL4, BMI1 and BMI1 [36]”. Please remove duplicates.

3. Page 5, 2nd paragraph: “Through bioinformatics analysis form the Cancer Genome Atlas”, should be “from”.

4. Page 7, 3rd paragraph: “Exosomes represent a kind of intracellular vesicles ...”. Is it “extracellular”?

TNM: please define.

5. Page 8: ref [90]: please make sure it is the right ref.

The middle section (e.g., %, specificity, sensitivity) is confusing; should be clarified or moved to Table 2.

6. Page 9, 2nd paragraph: regarding the ref [90], please double-check the descriptions in this paragraph and ensure this is the right article to cite.

7. Page 11, 1st paragraph: “89Y?” Please clarify.

3rd paragraph: “immune-affinity” should be immune-affinity?

Table 1: “miR-205-5p” needs one more citation [26].

“miR-214”: ref [32] is not appropriate.

Author's Response to Decision Letter for (RSOB-18-0212.R0)

See Appendix A.

Decision letter (RSOB-18-0212.R1)

02-Jan-2019

Dear Professor Liang

We are pleased to inform you that your manuscript entitled "Emerging microRNA biomarkers for colorectal cancer diagnosis and prognosis" has been accepted by the Editor for publication in Open Biology.

You can expect to receive a proof of your article from our Production office in due course, please

check your spam filter if you do not receive it within the next 10 working days. Please let us know if you are likely to be away from e-mail contact during this time.

Article processing charge

Please note that the article processing charge is immediately payable. A separate email will be sent out shortly to confirm the charge due. The preferred payment method is by credit card; however, other payment options are available.

Sincerely,

The Open Biology Team

mailto:openbiology@royalsociety.org

ditage Insights by clicking on the following link: <https://www.surveymonkey.com/r/author-perspectives-on-academic-publishing-royal-society>

This should take no more than 15 minutes and you will have the opportunity to enter a prize draw. We hope these results will provide us with valuable insights we can use to improve our service.

Appendix A

27 Dec, 2018

Shufang Liang, Ph.D & Professor
State Key Laboratory of Biotherapy and Cancer Center
West China Hospital, Sichuan University
Chengdu, Sichuan, P.R. China

Dear Editors,

On behalf of my co-authors, I would like to submit our revised paper entitled “Emerging microRNA biomarkers for colorectal cancer diagnosis and prognosis” for publication in *Open Biology*. According to the editorial comments, we dealt carefully with each of the points. Here, our point-by-point responses to the reviewers’ comments were integrated into the revised manuscript as enumerated below.

Referee 1

Comment 1. It would be better if the disadvantages of using microRNA as biomarkers are also summarized and discussed in the section subtitled “Advantages of microRNA biomarkers” on page 8.

Response: We agree with the reviewer’s good suggestion. In the revised version, we have added the concerns of disadvantages and specificity of microRNA biomarkers for CRC on page 9.

Comment 2. Is the list of microRNA biomarkers specific for CRC? Can any of them be applied to other types of cancers?

Response: Some predictive candidate miRNAs biomarkers are not specific for CRC, which are also reported to have diagnosis potentials for other kind of cancers. Fortunately, a panel of miRNAs can be used to distinguish CRC patients from healthy controls with a relative high sensitivity and specificity by testing in a large population of subjects. Therefore several miRNAs combinations are feasible to monitor cancer profiling. We have summarized and discussed the specificity of microRNA biomarkers for CRC on page 9.

Comment 3. Some sentences are not clear. For example: On the line 7, page 5, the sentence “Regarding to lymph node metastasis, although a previous evaluation about the correlations of miRNA expression patterns with CRC tumor stage by using miRNA microarrays showed no significant differences”. On the line 24, page 5, the sentence “Furthermore, miRNAs have great potential in CRC treatment and the overcome of resistance to cancer therapy ” is difficult to understand.

Response: We have carefully modified the language expression of the whole manuscript and made them easier to understand.

Comment 4. Abbreviations should be consistency in the manuscript. For example, the miRNA and microRNA. On the line 14, page 8, the form of the letter “p” ($p < 0.001$ and $P = 0.004$).

Response: We have carefully checked and corrected the abbreviations in the revised version.

Referee 2

Comment 1. Page 3 (Introduction), 3rd paragraph: “as promising biomarkers 8.” Need format revision.

Response: We are sorry for our carelessness. Now the mistake has been corrected.

Comment 2. Page 4 (Aberrant miRNA expressions and roles in CRC), 1st paragraph: please make sure [23] is the right reference; may include the MiR-7, 192/215 and 19b-1 into Table 1. 2nd paragraph: “... SALL4, BMI1 and BMI1 [36]”. Please remove duplicates.

Response: We are sorry for the previous error, and now we have corrected the ref[23] with a right one. We have added MiR-7, 192/215 and 19b-1 into Table 1 and removed the duplicated BMI1.

Comment 3. Page 5, 2nd paragraph: “Through bioinformatics analysis form the Cancer Genome Atlas”, should be “from”.

Response: We have revised these mistakes and supplemented the Table 1 in the latest manuscript.

Comment 4. Page 7, 3rd paragraph: “Exosomes represent a kind of intracellular vesicles ...”. Is it “extracellular”? TNM: please define.

Response: We are appreciated to the reviewer’s correction. And the abbreviation TNM has been defined on the Line 27 of Page 7.

Comment 5. Page 8: ref [90]: please make sure it is the right ref. The middle section (e.g., %, specificity, sensitivity) is confusing; should be clarified or moved to Table 2.

Response: The previous ref [90] has been checked and replaced with the current ref[108] in this revised version. And the relative content has been supplemented in the Table 2.

Comment 6. Page 9, 2nd paragraph: regarding the ref [90], please double-check the descriptions in this paragraph and ensure this is the right article to cite.

Response: The previous ref [90] has been checked and replaced with the current ref[108] in this revised version. The descriptions in this paragraph have been carefully checked, and the content here is derived from the right reference.

Comment 7. Page 11, 1st paragraph: “89Y?.” Please clarify.

3rd paragraph: “immune-affinity” should be immune-affinity?

Table 1: “miR-205-5p” needs one more citation [26].

“miR-214”: ref [32] is not appropriate.

Response: We are sorry for citing the wrong reference previously. We have replaced them with the right ones and updated the descriptions in the revised manuscript.